# Activity Prediction Based on Deep Learning Techniques

**Jinsoo Park** [1,*], **Chiyou Song** [2], **Mingi Kim** [2] **and Sungroul Kim** [2]

[1] Department of Industrial Cooperation, Soonchunhyang University, Asan 31538, Republic of Korea
[2] Department of ICT Environmental Health System, Graduate School, Soonchunhyang University, Asan 31538, Republic of Korea
* Correspondence: vtjinsoo@gmail.com; Tel.: +82-10-7303-2254

**Abstract:** Studies on real-time $PM_{2.5}$ concentrations per activity in microenvironments are gaining a lot of attention due to their considerable impact on health. These studies usually assume that information about human activity patterns in certain environments is known beforehand. However, if a person's activity pattern can be inferred reversely using environmental information, it can be easier to access the levels of $PM_{2.5}$ concentration that affect human health. This study collected the actual data necessary for this purpose and designed a deep learning algorithm that can infer human activity patterns reversely using the collected dataset. The dataset was collected based on a realistic scenario, which includes activity patterns in both indoor and outdoor environments. The deep learning models used include the well-known multilayer perception (MLP) model and a long short-term memory (LSTM) model. The performance of the designed deep learning algorithm was evaluated using training and test data. Simulation results showed that the LSTM model has a higher average test accuracy of more than 15% compared to the MLP model, and overall, we were able to achieve high accuracy of over 90% on average.

**Keywords:** human activity recognition; deep learning models; machine learning; MLP; LSTM

## 1. Introduction

Human activity and behavior recognition (HABR) is an area of research to gain a high level of knowledge about human activities from raw sensor inputs, and it is regarded as an essential component to realize the 4th industrial revolution. Technologies of the 4th Industrial Revolution society will primarily comprise IoT sensors broadly deployed almost everywhere and intelligent software technologies such as artificial intelligence, machine learning, and so on. These technologies are collectively used to make our daily lives better by making various useful applications viable, which include healthcare, surveillance, location-based services, silver-care services, etc. Let alone several of these interesting applications, environmental issues, including the air pollution problem, have drawn lots of attention, especially in most Asian countries where the economy is growing very rapidly. Many countries have made efforts to develop technologies to predict the level and impact of air pollution in our daily life so that people can avoid air pollution and protect themselves. Related research efforts have focused on predicting the level of air pollution based on data collected from the sensors deployed broadly with the aim of alerting the untargeted majority of people. This kind of service needs to be advanced to provide a personal care type of service so that each individual can have different environmental information. If this kind of service is desired, it is essential to track the precise movement of individuals so that it is possible to let people know of their level of pollution in advance. Thus, it is important to predict the activities or behaviors of humans in a very reliable manner. This paper strived to develop deep learning prediction models to predict the activities of humans, using datasets collected from body-worn sensors designed to obtain personal environmental data during daily life. Related studies have presented online and offline predictive technologies for human activities, primarily in outdoor or indoor environments. However, our daily life

is composed of lots of diverse indoor and outdoor activities; therefore, research efforts should develop into working in any environment afterward. In this study, we developed two machine-learning-based activity prediction models applicable to our daily lives and evaluated the performance of the models to weigh up the possibility of their practical use.

Lots of technologies have been proposed to predict human behaviors and activities. Most of them are classified into two different categories depending on the type of data collected [1]. We can predict human behaviors using image or video data (image-based technologies) or sensor data (sensor-type technologies) collected from mobile devices or from stationary sensors deployed at home or in any target area. For the literature review, we excluded image-based studies and focused primarily on those using non-image data, collected especially from mobile devices. Sensor data can be classified into two different categories, mobile or stationary. Mobile-type sensors are, in general, worn by the subject person and generate data associated with the movements of the subject person, so it is highly likely to contain a significant amount of missing data or noise caused by the movements [2–5]. Stationary sensors are, in general, installed in places of interest, and the collected data contains relatively fewer missing values and noise compared to those of mobiles [6–10]. It is also important to differentiate whether the activity recognition is performed in online or offline environments. Online recognition, in general, refers to approaches in which the recognition tasks occur mostly in local devices and are executed in real time. Offline recognition, on the other hand, works in client–server computing environments and does not require real-time processing.

Our designed classifiers are developed to apply to both environments and to work online as well as offline. Further, there are lots of classifiers to realize the prediction tasks using the collected data. Those classifiers include classic machine-learning-type classifiers as well as modern deep-learning-type classifiers. The classic machine learning types of classifiers include the naïve Bayes classifiers [2,11,12], decision trees [13], hidden Markov models [3,14,15], support vector machines [16] and etc. Modern machine learning algorithms include well-known convolutional neural networks (CNN) [17], recurrent neural networks (RNN) [18,19], long short-term memory (LSTM) [20–29], etc. These classifiers are data-driven classifiers that require labeled data to train the classifiers themselves.

As we described previously, we implement two classifiers for activity recognition using data collected from portable body-worn sensors to obtain personal environmental information. Most previous research primarily addressed activity recognition in either indoor or outdoor environments. However, our research attempted to predict activities in more heterogeneous environments, reflecting our various daily life in diverse environments. We assumed 13 different activities that are typical to Korean families and tried to predict the activities using well-known MLP and LSTM models. Simulation results showed that the LSTM model has higher accuracy compared to that of the MLP model.

This paper is organized as follows. The first section provides in-depth details on the collection of raw data, including the place where the data were collected and the subject persons who volunteered for the collection task. The second section provides technical details on the deep learning models used. The third section describes simulation results comparing the performance of the two deep learning models. Finally, the fourth section provides conclusions and insights from the results.

## 2. Materials

### 2.1. Data Collection in Study Area

Four volunteers living in Seoul, Kyunggi, Choongchung, and Jeolla provinces, South Korea, were chosen as subject persons who carry body-worn sensors to measure the concentration levels of fine dust and several other environmental data. The volunteers also recorded their activities for every hour, excluding their sleep time. Figure 1 shows a sample trajectory of a subject person who moved from Choongchung province to Seoul City.

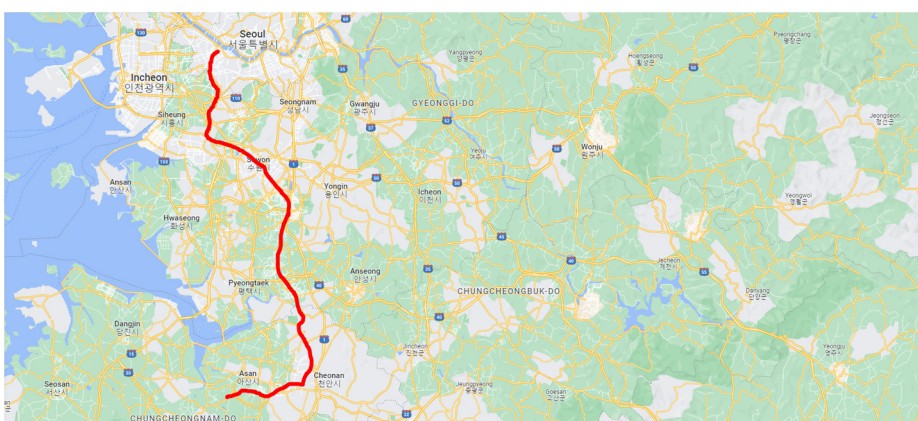

**Figure 1.** Sample trajectory (in red) of a subject person for a day.

The dataset was collected for 2 months, from 1 May 2018 to 31 June 2018, using a commercial sensor [30], and the sensor is capable of transmitting pressure, temperature, and humidity information through a WiFi connection along with air quality measurement. The dynamic range and accuracy of the sensor related to our dataset are given in Table 1.

**Table 1.** Sensor specifications.

| Specifications | Range |
|---|---|
| Temperature range | $-40\sim185\,^{\circ}\text{F}$ |
| Effective $PM_{2.5}$ range | $0\sim500\,\mu\text{g/m}^3$ ($\geq1000\,\mu\text{g/m}^3$ for max) |
| Accuracy tolerance for relative humidity | $\pm3\%$ |
| GPS accuracy | 2.5 m |
| Altitude range | <18,000 m |
| Velocity range | <515 m/s |

The dataset contains several environmental variables such as particulate matter 2.5 ($PM_{2.5}$), temperature (Temp), relative humidity (RH), and informative data related to human trajectory. The data were sampled every 2 min. Further, we pre-defined activity patterns into 13 different categories: Commuting with a bus, Commuting with a car, Commuting with a subway, Commuting with a train, Commuting with an elevator, Home-BBQ, Pan-Frying, Home-SHS, Staying inside home, Staying inside work place, Staying outside, Visiting other commercial place, Visiting restaurant-café, and Walking at outside. A snapshot of the dataset is given in Figure 2.

### 2.2. Details on the Dataset

Four subject persons were given IDs, ID_01, ID_02, ID_03, and ID_04, respectively. Figure 3 shows the overall plots of the environmental and trajectory-related data for the ID_4 during the whole observation period as an example. The plots include profiles of activities and environmental information measured while the subject person is on the move. However, the location, altitude, and speed information of the subject is omitted to focus more on the relationship between the activity pattern and the environmental information. Each piece of data was plotted with respect to the observation time when it was collected, and the chosen data were used as features for the classifier to predict the activity pattern for the next observation time. Activity patterns are enumerated for the convenience of simulation and visualization as follows: Commuting with a bus as 1, Commuting with a car as 2, Commuting with a subway as 3, Commuting with a train as 4, Commuting with an elevator as 5, Home-BBQ Pan-Frying as 6, Home-SHS as 7, Staying inside home as 8, Staying inside work place as 9, Staying outside as 10, Visiting other commercial place as 11, Visiting restaurant-café as 12, and Walking at outside as 13.

| 1 | LATITUDE | LONGITUDE | ALTITUDE | SPEED | Temp | RH | pm2_5 | activity13 |
|---|---|---|---|---|---|---|---|---|
| 2 | 36.769827 | 126.930517 | 86.922035 | 4.419826 | 83 | 32 | 39.08 | Staying inside work place |
| 3 | 36.773367 | 126.933531 | 90.560177 | 3.184919 | 81 | 35.5 | 35.01 | Walking at outside |
| 4 | 36.774142 | 126.933672 | 101.02568 | 2.597194 | 81 | 37 | 40.27 | Walking at outside |
| 5 | 36.775383 | 126.934123 | 104.630257 | 3.176346 | 81 | 39.66666667 | 51.78666667 | Walking at outside |
| 6 | 36.776141 | 126.934168 | 109.642675 | 1.558012 | 80 | 42 | 50.71 | Visiting restautant, cafe |
| 7 | 36.776135 | 126.933812 | 117.141511 | 1.432506 | 81.5 | 41.5 | 49.24 | Visiting restautant, cafe |
| 8 | 36.775862 | 126.933428 | 116.23637 | 0.620822 | 82 | 38 | 63.46 | Visiting restautant, cafe |
| 9 | 36.775894 | 126.933604 | 119.67766 | 1.135787 | 83 | 36 | 53.565 | Visiting restautant, cafe |
| 10 | 36.776003 | 126.934079 | 107.168112 | 1.140761 | 84 | 36 | 50.53 | Visiting restautant, cafe |
| 11 | 36.776022 | 126.93455 | 95.804507 | 1.055559 | 84 | 35 | 49.76 | Visiting restautant, cafe |
| 12 | 36.776104 | 126.934448 | 90.256293 | 0.933425 | 84 | 35 | 48.46 | Visiting restautant, cafe |
| 13 | 36.776118 | 126.934382 | 86.974403 | 1.377597 | 84.5 | 34.5 | 47.565 | Visiting restautant, cafe |
| 14 | 36.776156 | 126.934407 | 82.348356 | 0.879979 | 84 | 34 | 74.74 | Visiting restautant, cafe |
| 15 | 36.776178 | 126.934479 | 75.440788 | 0.807282 | 85 | 34 | 161.515 | Visiting restautant, cafe |
| 16 | 36.77623 | 126.934548 | 67.770783 | 1.132503 | 85 | 34 | 175.08 | Visiting restautant, cafe |

**Figure 2.** A snapshot of the dataset. It contains several environmental data such as temperature (Temp), relative humidity (RH), particulate matter2.5 (pm2_5), as well as GPS data (Latitude and Longitude), altitude, and speed data that relate to human trajectory, plus pre-defined labels for the 13 activity patterns (activity13).

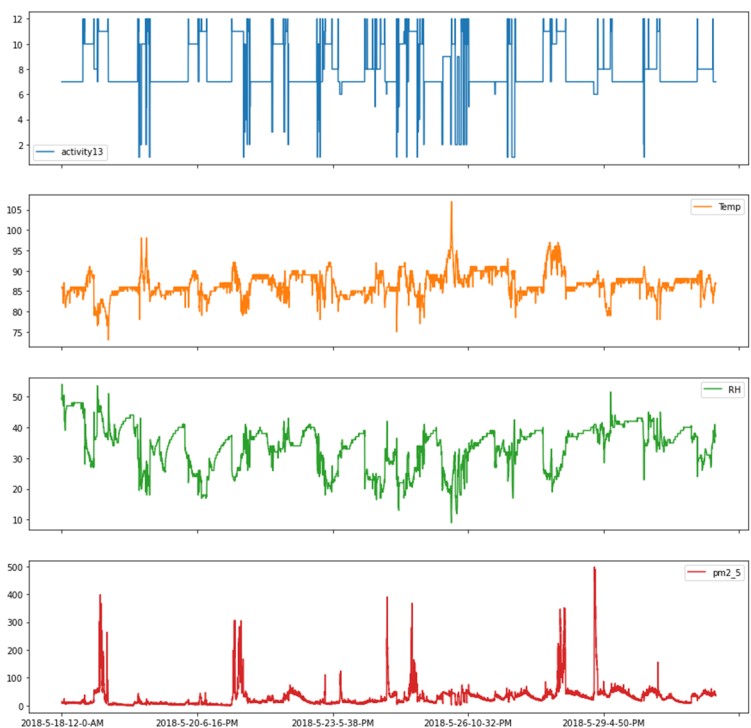

**Figure 3.** Sample data profiles during an observation period.

Samples of activity profiles are depicted in Figure 4, where activity patterns during the morning and evening periods are very dynamic compared to those of other time periods. As shown on the left, the activity pattern changes from 13 (Walking outside) to 1 (Commuting with a bus), which indicates that the subject moves from home to working place or other outdoor activities. Meanwhile, in the evening hours, most people come home after work or meet others outside, and corresponding activity patterns can be observed on the right-hand side of Figure 4. From Figure 4, we can conclude that the data collection

process was properly executed, and the collected data show a contextual connection for each activity.

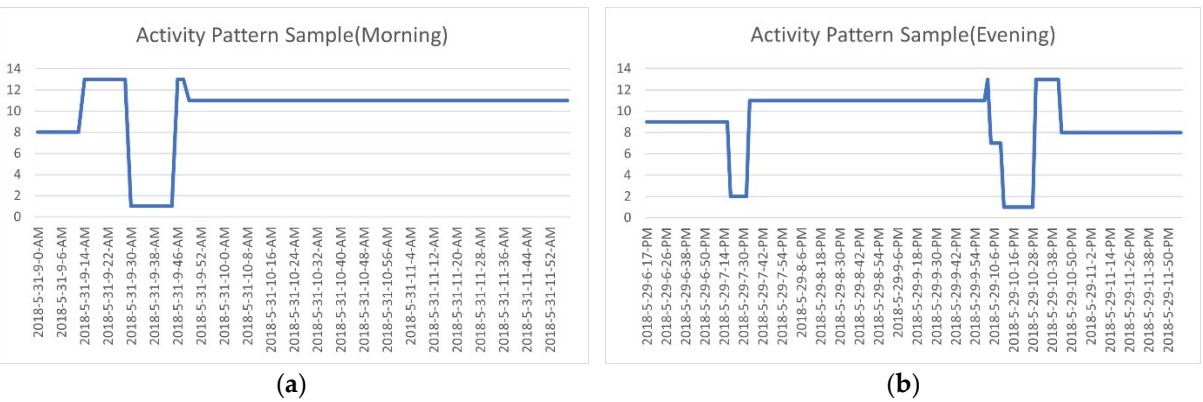

**Figure 4.** Samples of activity pattern profiles during the morning and evening periods: (**a**) an activity pattern profile during the morning and (**b**) an activity pattern profile during the evening.

### 2.3. Analysis of PM$_{2.5}$ Exposure and Activity Pattern

As described previously, we know that our daily activities are accompanied by fine dust particles with a diameter of less than 2.5 μm, which are hazardous to our health, especially for the elderly and children [31]. To investigate the relationship between the distribution of PM$_{2.5}$ and each activity, we investigated the average sojourn time of each subject's activity and the corresponding distribution of PM$_{2.5}$. Table 2 shows the number of activities performed by each subject during a day and the proportion (%) of corresponding activities (mean $\pm$ SD) that the subject spent for a day. The table shows that people spend more time in indoor-like environments than outdoor-like ones, indicating that fine dust distribution in indoor-like environments is more important to our health and requires careful study. Several activities were not observed at all, and the corresponding cases were omitted in Table 1. Table 3 shows the percentiles of PM$_{2.5}$ concentrations per each activity. It shows clearly that PM$_{2.5}$ concentration is high in indoor-like environments compared to outdoor environments. These tables imply that personalized environmental warning technologies are necessary to provide customized per-person environmental information. Machine-learning-based techniques can provide viable solutions for customized environmental services by predicting the next human activities properly.

**Table 2.** Average fraction of sojourn time (mean $\pm$ SD) that subject persons spent per day (%).

| ID | Commuting with a Bus | | Commuting with a Car | | Commuting with a Subway | | Staying inside Home | | Staying inside Work Place | |
|---|---|---|---|---|---|---|---|---|---|---|
| | N (day) | Mean $\pm$ SD | N (day) | N (day) | Mean $\pm$ SD | N (day) | Mean $\pm$ SD | Mean $\pm$ SD | N (day) | Mean $\pm$ SD |
| ID_01 | 5 | 72.0 $\pm$ 93.3 | 4 | 8 | 502.0 $\pm$ 215.0 | 3 | 96.7 $\pm$ 42.4 | 20.0 $\pm$ 19.0 | 1 | 20.0 $\pm$ NA |
| ID_02 | 12 | 40.3 $\pm$ 23.6 | 4 | 14 | 581.0 $\pm$ 167.0 | 5 | 368.0 $\pm$ 228 | 19.0 $\pm$ 4.76 | 11 | 35.5 $\pm$ 19.1 |
| ID_03 | 1 | 6.0 $\pm$ NA | 1 | 14 | 639.0 $\pm$ 110.0 | 6 | 296.0 $\pm$ 159.0 | 0.0 $\pm$ NA | 3 | 98.0 $\pm$ 68.4 |
| ID_04 | 6 | 26.0 $\pm$ 37.5 | 4 | 14 | 635.0 $\pm$ 159.0 | 6 | 313.0 $\pm$ 138.0 | 72.5 $\pm$ 91.1 | 4 | 30.5 $\pm$ 14.0 |
| Total | 15 | 69.1 $\pm$ 58.0 | 10 | 19 | 672.0 $\pm$ 98.8 | 7 | 433.0 $\pm$ 221.0 | 48.4 $\pm$ 57.7 | 14 | 55.4 $\pm$ 52.6 |

| | Staying Outside | | Visiting Other Commercial Place | | Visiting Restaurant-Café | | Walking at Outside | |
|---|---|---|---|---|---|---|---|---|
| | N (Day) | Mean $\pm$ SD | N (Day) | Mean $\pm$ SD | N (Day) | Mean $\pm$ SD | N (Day) | Mean $\pm$ SD |
| | 1 | 2.0 $\pm$ NA | 7 | 222.0 $\pm$ 177.0 | 6 | 58.3 $\pm$ 49.9 | 8 | 80.2 $\pm$ 40.5 |
| | 2 | 53.0 $\pm$ 66.5 | 7 | 42.0 $\pm$ 59.6 | 10 | 408.0 $\pm$ 311 | 13 | 78.3 $\pm$ 50.7 |
| | 1 | 144.0 $\pm$ NA | 8 | 87.8 $\pm$ 46.0 | 6 | 35.0 $\pm$ 30.7 | 11 | 77.6 $\pm$ 51.0 |
| | 2 | 135.0 $\pm$ 191.0 | 12 | 174.0 $\pm$ 107.0 | 11 | 167.0 $\pm$ 147.0 | 14 | 63.0 $\pm$ 48.5 |
| | 6 | 87.0 $\pm$ 108.0 | 18 | 239.0 $\pm$ 128.0 | 17 | 313.0 $\pm$ 278.0 | 19 | 158.0 $\pm$ 85.5 |

NA: Not available.

**Table 3.** Percentiles of PM$_{2.5}$ concentration per person and activity.

| ID | Commuting with a Bus | | | | Commuting with a Car | | | | Commuting with a Subway | | | | Commuting with a Train | | | | Commuting with an Elevator | | | | Home-BBQ Pan-Frying | | | |
|---|---|---|---|---|---|---|---|---|---|---|---|---|---|---|---|---|---|---|---|---|---|---|---|---|
| | N (Day) | 25% | 50% | 75% | N (Day) | 25% | 50% | 75% | N (day) | 25% | 50% | 75% | N (Day) | 25% | 50% | 75% | N (Day) | 25% | 50% | 75% | N (Day) | 25% | 50% | 75% |
| ID_01 | 5 | 12.7 | 18.9 | 44.6 | 4 | 31.3 | 35.6 | 39.4 | 1 | 17.7 | 20.2 | 23.3 | 0 | - | - | - | 0 | - | - | - | 2 | - | - | - |
| ID_02 | 12 | 7.8 | 11.6 | 36.2 | 4 | 19.8 | 30.6 | 42.0 | 11 | 12.8 | 19.9 | 35.5 | 0 | - | - | - | 0 | - | - | - | 2 | - | - | - |
| ID_03 | 1 | 4.6 | 10.7 | 17.3 | 1 | 50.3 | 50.3 | 50.3 | 3 | 22.8 | 36.8 | 42.4 | 0 | - | - | - | 0 | - | - | - | 5 | - | - | - |
| ID_04 | 6 | 11.8 | 25.1 | 32.9 | 4 | 4.1 | 9.0 | 18.0 | 4 | 16.0 | 32.4 | 34.9 | 3 | 12.2 | 26.3 | 40.6 | 0 | - | - | - | 4 | - | - | - |
| Total | 15 | 9.5 | 18.1 | 36.4 | 10 | 5.9 | 16.6 | 32.8 | 14 | 14.5 | 29.2 | 38.5 | 3 | 12.2 | 26.3 | 40.6 | 5 | - | - | - | 10 | - | - | - |

| Home-SHS | | | | Staying inside Home | | | | Staying inside Work Place | | | | Staying Outside | | | | Visiting other Commercial Place | | | | Visiting Restaurant-Café | | | | Walking at Outside | | | |
|---|---|---|---|---|---|---|---|---|---|---|---|---|---|---|---|---|---|---|---|---|---|---|---|---|---|---|---|---|
| N(Day) | 25% | 50% | 75% | N (Day) | 25% | 50% | 75% | N (Day) | 25% | 50% | 75% | N (Day) | 25% | 50% | 75% | N (Day) | 25% | 50% | 75% | N (Day) | 25% | 50% | 75% | N (Day) | 25% | 50% | 75% |
| 3 | 59.1 | 69.9 | 74.2 | 8 | 27.9 | 33.7 | 39.6 | 3 | 34.2 | 39.1 | 47.4 | 1 | 68.5 | 69.6 | 71.8 | 7 | 32.0 | 38.8 | 43.9 | 6 | 42.8 | 55.0 | 109.0 | 8 | 34.6 | 42.7 | 53.5 |
| 0 | - | - | - | 14 | 11.1 | 23.6 | 35.4 | 5 | 14.7 | 32.9 | 44.5 | 2 | 30.7 | 32.2 | 34.4 | 7 | 13.2 | 19.2 | 45.5 | 10 | 18.9 | 38.7 | 49.8 | 13 | 17.7 | 25.5 | 37.1 |
| 0 | - | - | - | 14 | 11.1 | 24.2 | 34.3 | 6 | 17.7 | 36.0 | 45.6 | 1 | 6.5 | 6.9 | 7.3 | 8 | 6.6 | 11.5 | 26.4 | 6 | 19.6 | 41.5 | 48.8 | 11 | 12.9 | 20.7 | 40.7 |
| 0 | - | - | - | 14 | 7.7 | 16.8 | 31.3 | 6 | 14.6 | 36.2 | 43.4 | 2 | 35.9 | 41.0 | 45.4 | 12 | 5.9 | 11.8 | 25.4 | 11 | 28.8 | 49.3 | 107.0 | 14 | 14.3 | 26.5 | 40.3 |
| 3 | 59.1 | 69.9 | 74.2 | 10 | 11.0 | 24.4 | 35.0 | 7 | 15.4 | 36.1 | 44,8 | 6 | 8.0 | 35.1 | 41.9 | 18 | 10.4 | 25.7 | 39.7 | 17 | 25.5 | 41.2 | 61.4 | 19 | 16.8 | 28.5 | 42.7 |

## 3. Design of Network Models

In this section, we describe the network structures of the chosen deep learning models in detail. The models include the well-known multilayer perceptron (MLP) and long short-term memory (LSTM) [25,32].

### 3.1. MLP Network Model Structure

The MLP is a multi-layered feedforward artificial neural network that maps input data sets to a set of appropriate outputs. It consists of multiple hidden layers, and nodes in the hidden layers are fully connected to the nodes in the following layer. The nodes of the layers are neurons with nonlinear activation functions, such as the sigmoid function, except for the nodes of the input layer. An MLP with one hidden layer can have the following structure, as shown in Figure 5. It consists of input nodes, output nodes, and hidden nodes, and the hidden nodes are linked to each of the input nodes and also to the output nodes with certain weights. This simple model can be extended to models with more hidden layers.

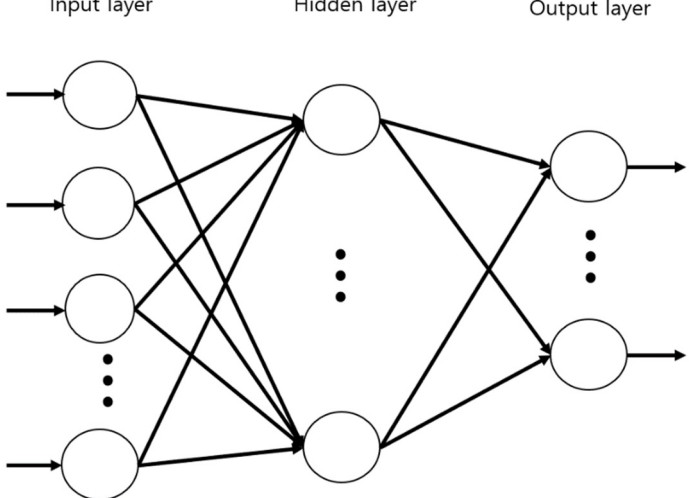

**Figure 5.** A structure of an MLP. Nodes at each layer are drawn in circles and the vertical dots indicate more nodes in each layer. And the nodes at each layer are connected by unidiretionallinks.

### 3.2. LSTM Network

#### 3.2.1. RNN Structure

Neural networks must have three properties to handle time-series data. First, it has to take features in an orderly fashion. Second, the length of the hidden layer has to be variable to handle variable lengths of data. Third, the model is capable of remembering the previous information, and can also use that at the time when it was requested. A recurrent neural

network (RNN) is a network model that was invented to predict time-series data, satisfying the previous three properties. It has a structure that is similar to that of MLP, but nodes in the hidden layer have edges to other hidden layer nodes, as shown in Figure 6. These edges are called "recurrent edges", which makes the network satisfy the three properties mentioned previously.

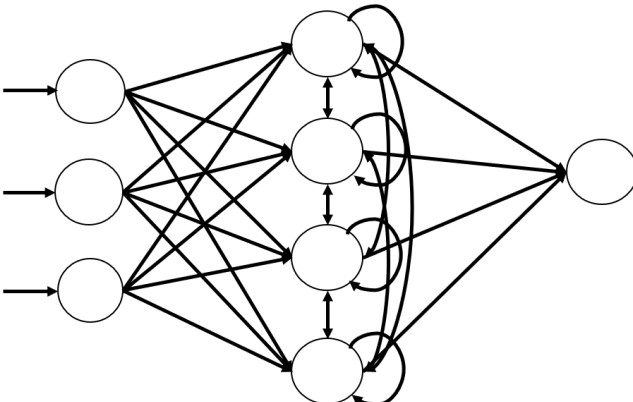

**Figure 6.** A sample structure of an RNN. There are three input nodes, four hidden layer nodes, and one output node. The nodes are interconnected with each other.

### 3.2.2. LSTM Structure

Due to structural drawbacks, RNN is not appropriate for predicting data with long-term dependence. Meanwhile, the LSTM, an advanced RNN model, has memory blocks in the hidden layer with several gates to handle data with long-term dependence, as shown in Figure 7. LSTMs deal with the long-term dependence issue by selectively controlling inputs and outputs with the gates. Since its introduction in 1995 [26], it has been advanced to include forget gates and peepholes, in addition to existing input and output gates. The peephole plays a role in notifying the states of the memory block to the three gates (yellow edges in Figure 7). It is very useful when a special action needs to be taken depending on a certain condition while processing time-series data.

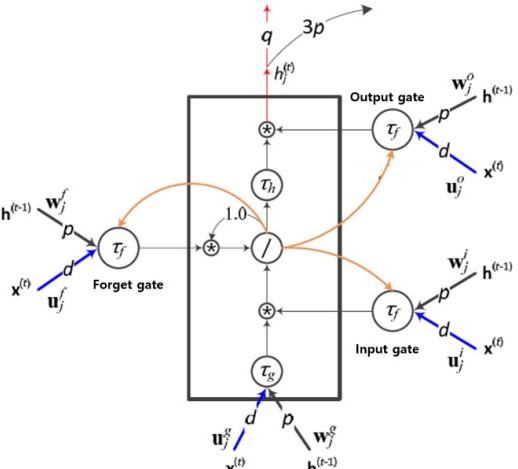

**Figure 7.** An LSTM memory block. This figure shows the jth memory block in a hidden layer. $x^{(t)}$ and $h^{(t-1)}$ represents input vectors at present state and previous hidden layer state, respectively. $\tau_g$, $\tau_h$, $\tau_f$ are the activation functions at the input, output, and the input and output gates, respectively. d, p, and q indicate the number of connections to the input nodes, hidden nodes, and output nodes, respectively. $w^g$, $w^i$, and $w^o$ are the weights for the recurrent edges connected to the inputs, input gates, and output gates, respectively. The symbols / and * refer to the linear activation function and multiplication respectively. In this figure, the biases are omitted for the simplicity of the diagram.

## 4. Results and Discussion

### 4.1. Activity Prediction

As described in Section 2, the recorded data contain longitudinal and latitudinal location, accelerometer, and elevation information, and they were recorded in two-minute periods. Figure 8 shows the overall procedure for classification tasks beginning with the preprocessing of data. The raw dataset contained a large number of missing values, and the data containing the missing values were excluded for simplicity of simulation in the preprocessing step. The preprocessing step includes the normalization of the dataset, for which z-score normalization is used [33]. If a proper imputation method is chosen for our data, the method can be used to impute the missing values for further experiments. The dataset was then separated into training (70%) and testing (30%). The 13 activity patterns are enumerated and one-hot coded for the simulation, as mentioned in Section 2. We used Keras, an open-source software library that provides a Python interface for artificial neural networks [34]. Both MLP and LSTM models were set up, as shown in Table 4. In the LSTM model, three previous data points with five features each (window size was set to 3) were used for the prediction of the data point at the next time stamp, so the input shape is $3 \times 5$. There is only one input and output layer, and the number of nodes at the hidden layer is chosen as 128. Both network models have 13 output nodes to match the 13 activity patterns. The activation functions chosen for the hidden layer and output layer were ReLU and SoftMax. The "Adam" was chosen as an optimizer [32].

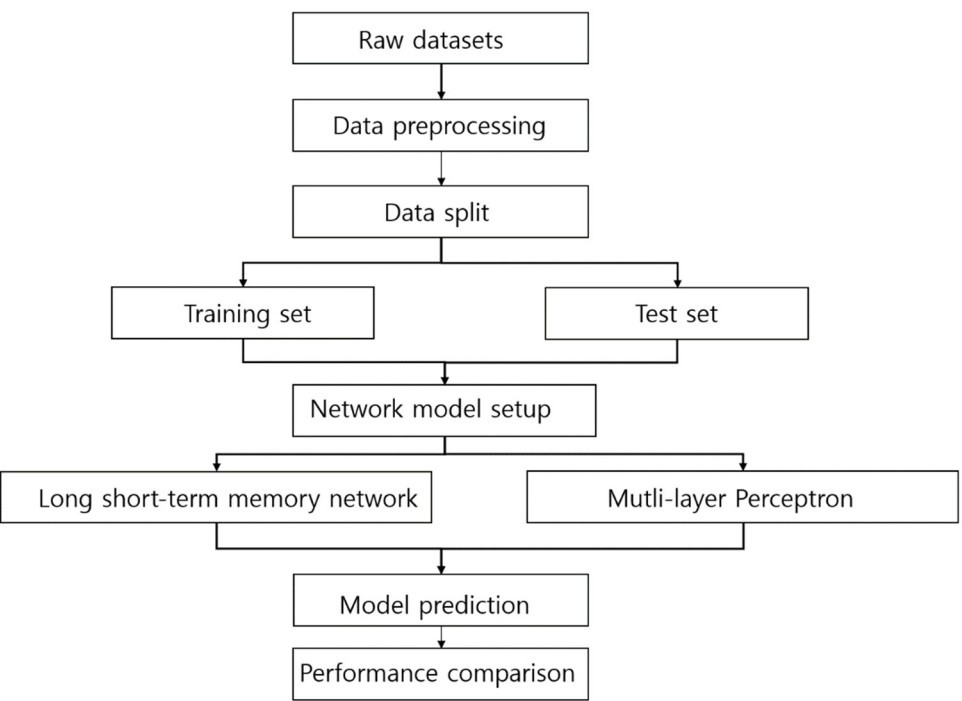

**Figure 8.** Activity prediction flow based on MLP and LSTM.

We performed an exhaustive search to determine the architecture of the two network models by changing the number of hidden layers from large values to small values, using sample training data as shown in Table 5. The table shows that the accuracy of LSTM significantly drops as the number of hidden layers increases, whereas MLP's accuracy stays relatively stagnant. Based on this pilot test, we chose the architecture of the models, as shown in Table 4. Moreover, we set the architecture and specification of the two models in close resemblance in order to run the simulations under similar environments.

**Table 4.** Network model setup for the MLP and the LSTM.

| Specifications | MLP | LSTM |
|---|---|---|
| Window size | 3 | 3 |
| Input dimension | $3 \times 5$ | $3 \times 5$ |
| Number of hidden layers | 1 | 1 |
| Number of out layers | 1 | 1 |
| Number of hidden nodes | 128 | 128 |
| Number of output nodes | 13 | 13 |
| Hidden layer activation functions | ReLU | ReLU |
| Optimizer | Adam | Adam |
| Output layer activation functions | SoftMax | SoftMax |

**Table 5.** Pilot training and test accuracy of ID_01 depending on a different number of hidden layers while fixing the number of hidden nodes.

| Number of Hidden Layers | Training Accuracy | | Test Accuracy | |
|---|---|---|---|---|
| | MLP | LSTM | MLP | LSTM |
| 10 | 0.98 | 0.57 | 0.94 | 0.53 |
| 5 | 0.99 | 0.59 | 0.94 | 0.55 |
| 1 | 0.97 | 0.99 | 0.94 | 0.96 |

*4.2. Performance Evaluation*

The performance of both models was evaluated in terms of predictive accuracy, as shown in Tables 6–9. For the experiments, we chose a learning rate of 0.01, epoch of 100, and batch size of 32, 64, 128, and 256, respectively. Training accuracy for both models was higher than the test accuracy for all subject persons. Further, for all subject persons, the training accuracy was more than 90%, except for the ID_01 case, where the test accuracy was lower than 90%. However, in general, the accuracy rates for all subject experiments were very high, which is more than 90%, except in the ID_01 case.

**Table 6.** Training and test accuracy of ID_01.

| Batch Size | Training Accuracy | | Test Accuracy | |
|---|---|---|---|---|
| | MLP | LSTM | MLP | LSTM |
| 32 | 0.97 | 0.98 | 0.83 | 0.85 |
| 64 | 0.97 | 0.98 | 0.83 | 0.86 |
| 128 | 0.96 | 0.98 | 0.80 | 0.86 |
| 256 | 0.95 | 0.97 | 0.77 | 0.86 |

**Table 7.** Training and test accuracy of ID_02.

| Batch Size | Training Accuracy | | Test Accuracy | |
|---|---|---|---|---|
| | MLP | LSTM | MLP | LSTM |
| 32 | 0.98 | 0.99 | 0.94 | 0.94 |
| 64 | 0.98 | 0.99 | 0.94 | 0.94 |
| 128 | 0.97 | 0.99 | 0.94 | 0.94 |
| 256 | 0.97 | 0.98 | 0.93 | 0.94 |

**Table 8.** Training and test accuracy of ID_03.

| Batch Size | Training Accuracy | | Test Accuracy | |
|---|---|---|---|---|
| | MLP | LSTM | MLP | LSTM |
| 32 | 0.97 | 0.99 | 0.97 | 0.98 |
| 64 | 0.97 | 0.98 | 0.96 | 0.98 |
| 128 | 0.96 | 0.98 | 0.96 | 0.98 |
| 256 | 0.95 | 0.98 | 0.95 | 0.97 |

**Table 9.** Training and test accuracy of ID_04.

| Batch Size | Training Accuracy | | Test Accuracy | |
|---|---|---|---|---|
| | MLP | LSTM | MLP | LSTM |
| 32 | 0.97 | 0.98 | 0.97 | 0.97 |
| 64 | 0.96 | 0.97 | 0.97 | 0.97 |
| 128 | 0.95 | 0.97 | 0.97 | 0.97 |
| 256 | 0.91 | 0.97 | 0.77 | 0.97 |

As shown in the table, the LSTM had higher accuracy for all batch-size experiments compared to the MLP. The test accuracy for both models was lower compared to that of the training data, which is expected in machine learning. However, given the heterogeneous nature of the collected data, i.e., both indoor and outdoor activity patterns, this difference was low and was not statistically significant. Nevertheless, our model may demonstrate even higher performance in more homogeneous environmental settings. Specifically, we anticipate that applying our model in such settings will result in improved performance compared with the current result.

It is worth examining the overall percentages of correct predictions for each activity pattern, as illustrated in Figure 9 below. The *x*-axis (x-label) in Figure 9 corresponds to each activity pattern, while the *y*-axis (y-label) indicates the ratio between the total number of occurrences and the number of correct predictions for each activity pattern. If there are no occurrences of activities, we omit corresponding drawings in the bar plot. This may correspond to the cases where a subject did not engage in any activities or when data collection was in error due to a sudden movement of the subject. The ratio was calculated for both training and test datasets. Overall, the training experiment had higher ratios compared to the test experiment. In the training data experiment, most activity patterns were correctly estimated, and activities involved with indoor environments had relatively higher ratios compared to outdoor activity cases. As shown in Figure 9, the ratios of correct predictions for each activity pattern are almost 1, indicating nearly 100% predictive accuracy. However, the test cases showed slightly lower ratios compared to the training cases. Among the test scenarios, activities associated with indoor-like environments, such as "Staying inside work place", "Visiting other commercial place", and "visiting restaurants, cafe" showed higher ratios compared to those of outdoor-like environments, except "Home-BBQ" and "Pan-Frying". In general, activities such as indoor cooking are the main cause of household air pollution and a leading environmental risk factor [35]. Sometimes, the PM2.5 concentration level of the cooking activities has a spiky distribution, resembling outliers, which could explain the corresponding lower accuracy in cooking activities. There were no data corresponding to Commuting with a train, Home-SHS, so there were no bars for these activities.

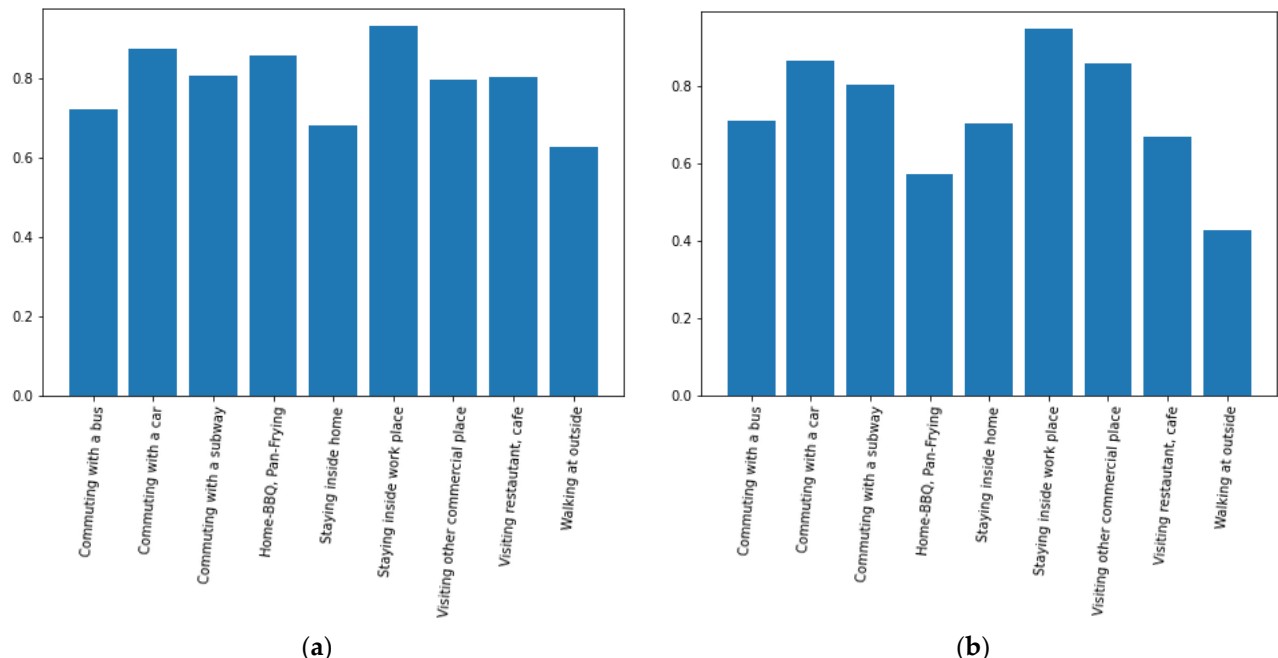

**Figure 9.** Prediction accuracy for each activity pattern: (**a**) for the training data; (**b**) for the test data.

As previously stated, our results clearly demonstrate that the predictive accuracy of our models can be improved if they are applied to homogeneous environmental settings. Additionally, we can further improve the performance of our current model if we elaborate more on behavior modeling as proposed by authors in [20] to predict the subject's next actions and collect data corresponding to the behavior model. The authors viewed human behaviors as a large collection of actions, activities, intra-activity behaviors, and inter-activity behaviors. Actions are defined as the simplest concepts of conscious muscular movements made by the subject. Activities are composed of several actions (e.g., taking a shower, watching a movie, etc.). The intra-activity behaviors describe how the subject performs a single activity at different times. The inter-activity behavior describes the chain of the subject's different activities. This issue will be included in our future studies.

One noticeable observation from the experiment is that the temperature and relative humidity did not play a significant role in the performance improvement, as shown in Table 10, presumably because the two features did not change significantly during the observation period. Table 10 highlights the change in the test accuracy of both models as more features were added.

**Table 10.** Training and test accuracy depend on different combinations of features. Case 1 corresponds to experiments performed using $PM_{2.5}$ feature only. Case 2 and 3 correspond to that of using $PM_{2.5}$ and RH and that of using $PM_{2.5}$, RH, and Temp, respectively.

| | | LSTM | | | MLP | |
|---|---|---|---|---|---|---|
| | | **Training Accuracy** | **Test Accuracy** | | **Training Accuracy** | **Test Accuracy** |
| | Case 1 | 0.99 | 0.9 | Case 1 | 0.97 | 0.81 |
| **ID_1** | Case 2 | 0.99 | 0.78 | Case 2 | 0.98 | 0.78 |
| | Case 3 | 0.99 | 0.74 | Case 3 | 0.98 | 0.73 |
| | Case 1 | 0.99 | 0.9 | Case 1 | 0.97 | 0.81 |
| **ID_2** | Case 2 | 0.99 | 0.78 | Case 2 | 0.98 | 0.78 |
| | Case 3 | 0.99 | 0.74 | Case 3 | 0.98 | 0.73 |

**Table 10.** *Cont.*

| | | LSTM | | | MLP | |
|---|---|---|---|---|---|---|
| | | Training Accuracy | Test Accuracy | | Training Accuracy | Test Accuracy |
| **ID_3** | Case 1 | 0.99 | 0.9 | Case 1 | 0.97 | 0.81 |
| | Case 2 | 0.99 | 0.78 | Case 2 | 0.98 | 0.78 |
| | Case 3 | 0.99 | 0.74 | Case 3 | 0.98 | 0.73 |
| **ID_4** | Case 1 | 0.99 | 0.9 | Case 1 | 0.97 | 0.81 |
| | Case 2 | 0.99 | 0.78 | Case 2 | 0.98 | 0.78 |
| | Case 3 | 0.99 | 0.74 | Case 3 | 0.98 | 0.73 |

## 5. Conclusions

This paper strived to infer activity patterns in both indoor and outdoor environments using environmental information, which distinguishes it from most previous studies that focus on the levels of $PM_{2.5}$ concentration affecting human health. We used a commercial multipurpose sensor to collect the raw data and designed deep learning models to infer the activity patterns using the collected raw data. We chose both MLP and LSTM network models for this research. MLP, a popular model in the 1980s, has recently gained interest again today due to the success of deep learning techniques in various applications, including speech recognition, image recognition, machine translation, etc. LSTM is a deep learning model that characterizes itself to handle a time-series type of data and has proved its characteristic adaptability for various applications such as voice recognition, stock index prediction, weather forecast, etc. During the performance comparison, we found that LSTM outperformed MLP in terms of prediction accuracy, which was expected considering the nature of the LSTM. More specifically, the accuracy was higher in the indoor-like environments than in the outdoor-like environments in both training and test simulations, except for the case of indoor cooking activities. Moreover, considering that all four features used for this research could be unstable under outdoor environments compared to indoor environments, we believe that the test accuracy of around 90% is very high. However, LSTM took significant amounts of computation time compared to MLP due to the complexity of network architecture, especially when the number of hidden layers is more than two. Therefore, we aimed to reduce the number of hidden layers to make the model more practical.

In the current study, we acknowledge the lack of large differences in lifestyles of subjects for various reasons, which could have led to degradation of performance. This issue will be included in our next research. Additionally, we will try to apply our model to more diverse heterogeneous environments based on the results of this research and enhance the performance of the current models. For this purpose, it is worth building a conceptual model as in [21] and collecting datasets corresponding to the model for more precise activity prediction. In addition, one of the ways to enhance the performance of the models is to incorporate other deep learning architectures, such as CNNs, into current LSTM architectures. Certain parts of the time series data show very frequent local and temporal changes of movements, which may be suitable for CNN to capture. If CNN is used together with LSTM, the current performance can be further improved.

**Author Contributions:** Conceptualization, J.P. and S.K.; methodology, J.P. and S.K.; software, J.P.; validation, J.P., C.S., M.K. and S.K.; data analysis, J.P., C.S., M.K. and S.K.; resources, J.P., C.S., M.K. and S.K.; data curation, J.P. and S.K.; writing—review and editing, J.P. and S.K.; visualization, J.P.; project administration, S.K.; funding acquisition, S.K. All authors have read and agreed to the published version of the manuscript.

**Funding:** This study was funded by Environmental Health Research Center Project (2016001360002) by the Korea Environmental Industry and Technology Institute, Ministry of Environment, South Korea.

**Institutional Review Board Statement:** Not applicable.

**Informed Consent Statement:** Not applicable.

**Data Availability Statement:** The data presented in this study are available on request from the corresponding author. The data are not publicly available due to research contract.

**Acknowledgments:** Authors thank study participants and their parents. This study was supported by the Soonchunhyang University Research Fund and the Brain Korea (BK) 21 (Big Data Analytics for Air Pollution and Health, ICT Environmental Health System, Graduate School, Soonchunhyang University).

**Conflicts of Interest:** The authors declare no conflict of interest.

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
