# Peer review of "Activity Prediction Based on Deep Learning Techniques"

_applsci, doi:10.3390/app13095684_

Round 1

Reviewer 1 Report

This paper describes attempts to recognize human activities in more realistic environments using deep-learning network models such as multilayer perception model and a long-short term memory(LSTM) model.

The article deals with the issues of forecasting and analyzing the level of air pollution in our daily life based on the analysis of data obtained using IoT technologies. The research work can be considered original and relevant. The paper is easy to read and follow. The methodology and results are well presented. However, the current content should be improved before publication.

1.      It is appropriate to compare the results of Table 4 presented in the paper with some classic machine learning classifiers.

2.      The data set should be divided into three sections (training, validation and testing).

3.      More details should be provided on what normalization procedures were performed on the data set.

4.      The research work should provide detailed information on how the hyperparameters of the neural network architecture were chosen and what methods of regularization were used.

5.      Tables 4, 5, 6, 7 show the batch size in various sizes. It would be appropriate to describe why the experiments were carried out at different sizes.

6.      Discuss the limitations of your research.

Reviewer 2 Report

1. Why were only 13 actions chosen? Why not 9 or 20? The choice needs to be substantiated.

2. Is the relationship between human behavior and the environment not fully understood?

3. Why were only 4 respondents selected for data collection? For example, not 5, 6 or 15. The choice of 4 respondents must be justified.

4. There are no criteria for selecting respondents. For example, gender, age, etc.

5. Data collection was carried out for 2 months. Is this period enough to solve the task? It needs to be substantiated.

6. 158-159 strings: “…These two tables insinuate…” is not a scientific style of presentation.

7. In general, in Section 2, the data are not sufficiently prepared to proceed to the use of the MLP and LSTM methods.

8. The well-known MLP method is described in great detail. It is advisable to significantly shorten the description of the MLP method.

9. The obtained results of the calculations are quite high, since the behavior of the respondents was very similar. For example, if the respondents lead different lifestyles, for example, a sportsman, an employee, a pensioner, or a mother of many children. Then the results may be different.

Reviewer 3 Report

The technical novelty of this paper is very weak. The authors have explained the proposed methods in section 3 with less novelty. The paper needs huge improvment.

Reviewer 4 Report

The paper presents Deep Learning algorithms for human activity recognition. For this purpose, it uses a dataset created to collect data on the activities of four people in a metropolitan area and its surroundings. It then applies two types of neural networks: MLP and LSTMs.

From the point of view of scientific interest, the construction of the dataset is the most significant aspect of the paper since the use of only two deep learning algorithms is not justified. It is deduced from the paper that the number of data is not especially large (this is where deep networks make sense to be used), and therefore, it is more relevant to start with more classical models. These are discussed in the paper (Naive, SVM etc.) but are neither applied nor compared with the references indicated. The latter is logical because the data used differs from the one developed in the paper.

The data taken are sequential (time series), and it does not make sense to use classical algorithms that do not take into account this temporal structure. MLP is used to introduce temporal sequences of three samples at a time (the authors call it window, which makes sense to use in LSTM but not in MLP), so what is really provided is a classifier that takes into account the last 6 minutes (it is assumed according to the authors that each data sample is taken in two minutes) to predict the following sample (prediction based on data from 6 minutes before). The deep networks that contemplate time series prediction (forecasting) should have been compared: RNN, LSTM and GRU. Moreover, since it seems to be a small amount of data (the authors do not indicate the size of these data nor the number of samples), ARIMA-type algorithms (SARIMA, etc.) should be tested first.

Specific comments:

It is stated in the paper that the data collected has no assigned values (N/A). In the case of GPS, it is extremely rare not to have some samples without values because they depend on satellite or Wifi coverage (it is common in areas without coverage, such as the subway in specific depths and some indoor areas due to the structure of the buildings). It would be good to clarify which sensor technologies are used for the data and to clarify this point.

Lines 68-69. It is said that classifiers have been developed to work online or offline. What does this mean? Is there an online version (API?)? Has the predictor associated with the classifier been integrated with the data collection application? Clarify this.

Line 152. Change redaction: "Table 1 shows...

Results and discussion section

-The specificities of the two networks are presented, but the reason for the size of the inner layer (128) is not justified. Nor have tests been done with more types of structures (more layers and or more cells/nodes per layer).

- The tests have only used one hyperparameter (Batch size). This parameter is closely associated with the input data (processing in memory of that batch size). As the size is 5 data x 3 samples = 15, with small batch sizes, it is expected that it works with better accuracy (as can be seen in Tables 4,5,6 and 7). The hyperparameter space should be better worked out, and other non-linear activation functions (here only Relu is used), number of nodes per layer or number of layers should be considered. In the case of LSTMs, a multiple of number (x1,x2) of the temporal input "window" considered is usually used (here, it would be for example, three layers).

Line 274. It is commented that the results are much better in indoor activities than in outdoor activities. From the profiles presented in the data, it seems logical that this is because people tend to spend more time in indoor areas than in movement, so in the samples, there is more than likely an evident bias towards indoor activities. This should be corrected in some way, and the easiest way to do it is to subsample (taking part of the samples, in intervals longer than two minutes, for example, 10 or more).

For all the above, that whole part of the paper should be redone, and since it is time-consuming work, it is recommended to reject the paper and submit it again with this work done.

Reviewer 5 Report

This paper demonstrates to use neural network models, MLP and LSTM, and perform human activity recognition. As far as I know, there have been many existing works to perform such a task or similar tasks, which are generally time series classification, so the novelty of this work is limited. It is great that the data is collected from actual subject persons in actual scenarios, which makes the results more convincible. But my greatest concern is that the novelty is still generally weak. I would suggest the authors to include more details about how the data is collected, and how these features are selected for this specific task, which I think would be better to show the novel works that the authors have done. My detailed comments are as follows:

1.      Sec. 1, the authors indicate the problem of the existing works is that they recognize human activity in either outdoor or indoor environments. However, what is the major difference to perform this task in outdoor and indoor environments? It is not clear why a previous work that can accurately recognize certain activity in outdoor environment cannot do well in indoor environment, unless the authors can list some related works to prove this and explain it in the manuscript. I would suggest the authors to further include some experiment results to prove it.

2.      Sec. 2, it is interesting to collect data from actual subject persons, but I would expect more details about it. Specifically, are the body-worn sensors designed and fabricated by the authors, or they are commercial products from a certain vendor? In either way, the authors should include more details about the sensors, in order to make this experiment reproduceable. In addition, some features are selected, such as temperature, humidity, pm2.5, etc., but the authors need to explain why these features are specifically selected for this task? Which one or ones would be the most useful in order to ensure accurate recognition? This should be evaluated in additional experiments. Finally, why these 13 activities? Will the result be different if someone use the neural network models of this work to recognize other activities?

3.      Sec. 3, I think the title of this section is misleading, as MLP and LSTM are not really proposed but used for the task.

4.      Table 3, please explain how are the architectures and the specifications of the models determined? How would the result and the conclusion be different if a MLP or LSTM with different specifications is used?

5.      Sec. 4.2, it is not surprised at all that LSTM results in better accuracy on test set, compared with MLP, as the human activities are generally time series data. Any new findings can the authors reveal from the results?

6.      Sec. 4, I would suggest the authors to include some experiment results by using some traditional machine learning algorithms (SVM, ARIMA, etc.), and compare the results to the ones produced by neural network models. This is to reveal whether a neural network model, which generally has greater computational time for either training and inference than traditional machine learning algorithms, would be necessary for such a task.

Reviewer 6 Report

The process of interpreting human motion using computer and machine vision technology is known as human activity recognition (HAR). Human motion can be defined as activities, gestures, or behaviors captured by sensors. After that, the movement data is converted into action commands that computers can use to execute and analyze human activity recognition code. The authors used a well-known multilayer perception(MLP) model and a long-short term memory(LSTM) model to recognize the human activities under more realistic environments. The article's originality is lacking; the study was the application of known methods in another field. Strenghs -> The methods used in the article are thoroughly described, including data collection and analysis techniques -> The validity and reliability of the methods used are also clearly established. -> The findings are effectively supported by graphs, tables, and diagrams and are easy to understand. Some improvements should be made to the article. -> Carefully check all grammatical error. Still, the English language should be improved. I suggest asking for help from a native English -> The abstract must be make strong. Abstract should be reviewed again. -> The Conclusions should be reviewed again. The original aspect of the study and its difference from other studies should be clearly explained. (The conclusion should be explored better and it needs to contemplate the eventual restrictions of the developed technique to address future works in this area.) -> It has been a comprehensive study in the literature in recent years. If there are more current literature studies, these should be examined in detail and added to the literature section (Especially, human activity recognition-identification studies.). It is a suggestion for the literature part of the article to be more comprehensive. It would be beneficial to review the following articles related to the subject and include them in the article. 1. “LSTM-CNN Architecture for Human Activity Recognition” (https://doi.org/10.1109/ACCESS.2020.2982225) 2. “A new approach for physical human activity recognition from sensor signals based on motif patterns and long-short term memory” (https://doi.org/10.1016/j.bspc.2022.103963) 3. “A multibranch CNN-BiLSTM model for human activity recognition using wearable sensor data” (https://doi.org/10.1007/s00371-021-02283-3) 4. "LSTM Networks Using Smartphone Data for Sensor-Based Human Activity Recognition in Smart Homes" (https://doi.org/10.3390/s21051636)

Round 2

Reviewer 3 Report

My comments on the paper titled “Activity Prediction Based on Deep Learning Techniques”

1. List out the main contributions of this work.

2. At the end of related work, please make a comparative Characteristic table which shows how your proposed method works well when compare to the traditional methods.

3. Please draw an in-depth diagram of your proposed method.

4. Is there any possibility to write a technical pseudo code for the proposed method (section 3)?

5. Please take metrics like log loss, f1 score, sensitivity, ROC, MRR, DCG, NDCG, IoU, Laplacian mean squared error (LMSE) and normalized absolute error (NAE) to evaluate the efficiency of the proposed approach.

I recommend major revision

Reviewer 4 Report

Regarding the comments

"We are not interested in sensor-related technologies, which is out of the scope of this research. We just used a commercial sensor from PurpleAir IncInc. Thank you."

Data quality is essential, so sensor technologies are important in this case. A paragraph in Section 2.1 must be included, which must comment on some features of the sensors (data precision for GPS and the rest of the sensors). 

Also, it is mandatory to check data quality by adding additional data about the dataset: na points presented (%) and the number of outliers.

"We used Keras, an open-source software library that provides a Python interface for artificial neural networks. There are several ways to determine the architectures and the specifications of the model: trial and error, heuristic search, exhaustive search, pruning and constructive algorithms, and etc. Our algorithm is close to exhaustive search. We tested two models by changing the number of layers and nodes in the hidden layers from large values to small values, using sample training data, and check what combinations output the best classification results. The best combinations are chosen as shown in Table 3. Also, we tried to have the architectures and the specifications of the models as similar as possible in order to run the simulations under similar environments and minimize the size of the model considering the practical implementation. Thank you so much. "

This comment must be added prior to the final used structure for MLP and LSTM. If you did those experiments, they must be referenced/explained before to clarify why the best structure was selected for the MLP and LSTM networks. So, it is mandatory to add a table with the different designs for MLP/LSTM networks (number of layers, etc.) and the accuracy obtained in these experiments.

Reviewer 5 Report

It looks the authors mixed the cover letters: the cover letter that I received is not really targeting to my comments, but to another reviewer. Thus, I cannot clearly see whether my previous comments have been properly addressed. I can only give the same decison as I have previously given.

Reviewer 6 Report

I think the article can be accepted in its present form.

Author Response

Thank you so much, sir.

Round 3

Reviewer 3 Report

The revised paper can be accepted now for the publication.

Author Response

Thank you so much.

JinSoo Park

Reviewer 5 Report

Some of my previous comments have been addressed in the revised paper. The authors also provided some explanations in their cover letter, where I think some explanations reveal some interesting findings, but it seems these explanations are missing in the manuscript. I would suggest the authors further include these explanations into the manuscript, such as:

1.   The authors claim “temperature and humidity did not play a significant role in the performance improvement because temperature and humidity level does not change significantly during observation period.” This is an interesting finding. Please include it in the manuscript. Also, can the authors prove it from the experimental results that they give in the paper?

2.   The authors claim “In general, the deep learning models outperform most classic machine learning algorithms as cited in most previous research.”, which is why the authors did not compare the performance of the neural network models with that of these traditional algorithms. However, just as the authors have claimed, deep learning models outperform the classical ML algorithms, “in general”, so there is no guarantee that deep learning models would win in such a specific task. If the authors cannot include additional experimental results to prove it, at least they should include some references where other researchers have proved deep learning is more suitable to be used for the certain task of activity prediction.

Round 4

Reviewer 5 Report

All of my previous comments have been properly addressed